# A fish cartel for Africa

Gabriel Englander [1] ✉ & Christopher Costello [2]

Many countries sell fishing rights to foreign nations and fishers. Although African coastal waters are among the world's most biologically rich, African countries earn much less than their peers from selling access to foreign fishers. African countries sell fishing access individually (in contrast to some Pacific countries who sell access as a bloc). We develop a bilateral oligopoly model to simulate the effects of an African fish cartel. The model shows that wielding market power entails both ecological and economic dimensions. Africa would substantially restrict access catch, which raises biomass by 16%. But this also confers economic benefits to all African nations, raising profits by an average of 23%. These benefits arise because market power shifts from foreign buyers to African sellers. While impediments to sustainable development like corruption are hard to change in the medium-term, deeper African integration is an already-emerging solution to African countries' economic and ecological challenges.

For centuries, rich countries appropriated natural resources from the less-rich countries they colonized. These countries have been independent for decades, but in many cases, the flow of raw materials from formerly-colonized countries persists. Some of these countries are poor, and for many of them, exporting natural resources is their primary source of tax revenue[1]. Fisheries may be the most poignant example; newly available satellite data reveal that foreign vessels, including those from more recently-industrialized China, Taiwan, and South Korea, comprise more than half of fishing activity in African waters and pay pennies on the dollar for access to these waters[2,3]. This state of affairs raises three questions: (1) Why are fishing agreements so apparently disadvantageous to the African countries that sign them?; (2) If African countries sold access as a cartel, as is done in other regions, how would they benefit economically?; and (3) What would be the conservation implications of such an African fish cartel?

While these questions may apply across a wide range of natural resources, such as critical minerals, oil, and natural gas, they are particularly acute in the case of international fishing access agreements. Many low- and middle-income countries sell the right to fish in their waters to high-income countries. These access agreements specify the allowable quantities, target species, and fishing methods (e.g., purse seine), as well as the access fees to be paid by foreign vessels or governments. For example, in 2019, Senegal allowed 28 European Union (EU) purse seine vessels to catch 6475 tons of tuna inside Senegal's waters. For that permission, Senegal received an access fee of $90 per

ton caught, and the EU vessels earned almost 20 times that amount ($1687 per ton) at the dock[4,5].

Selling fishing access rights is not unique to Africa. For example, Pacific Island countries like Kiribati earn nearly 50% of GDP from the sale of fishing access rights to foreign vessels. However, unlike African countries, who sell access individually (e.g., Senegal sells to the EU, and separately, Morocco sells to the EU), the nine Pacific island countries comprising the Parties to the Nauru Agreement (PNA) sell access to their waters jointly, acting as a cartel or bloc. Each year, PNA countries set a cap on the total quantity of tuna purse seine fishing in their collective waters and distribute a portion of that cap to each member country[6]. In 2019, PNA countries sold their portions of the cap for $454 per ton caught. PNA countries sell a nearly identical product as Senegal (the right to fish for tuna), yet their access fee is five times greater than Senegal's, suggesting that the per-ton profit they earn is much greater as well.

This pattern holds across African countries[7]. African countries earned an average of $128 per ton between 2010 and 2021, while PNA countries earned $307 per ton. Why do African countries earn so much less than PNA countries?

There are many potential explanations, including differences in enforcement against illegal fishing, biological differences in fish stocks, legal fishing costs, and corruption. If African countries enforce less than PNA countries on average, then this raises the expected value of fishing illegally in African waters, and lowers the value of signing a

[1]Development Research Group, The World Bank, Washington, DC, USA. [2]Environmental Markets Lab, Bren School of Environmental Science and Management, and Marine Science Institute, University of California, Santa Barbara, CA, USA. ✉e-mail: aenglander@worldbank.org

legal fishing agreement[8]. Similarly, if African fish stocks are depleted due to inferior fisheries management, are more likely to cross international boundaries, are composed of species that command lower market prices, or must be transported longer distances to final markets, then foreign willingness to pay to fish in African waters will also be lower[9]. For instance, tuna species, which generally command high market prices, account for only 26% of catch under public African access agreements, compared to a full 100% in PNA access agreements. Finally, if officials demand a bribe during negotiations in exchange for accepting a lower access fee on behalf of their country, then we will observe relatively lower access fees in African countries if they are more corrupt on average. Each of these offers a possible partial explanation for the large gap between access fees observed in the Pacific compared to access fees observed in Africa.

These potential causes of low African access fees, which suggest low profits, are difficult or impossible for African countries to change in the medium-term. However, another major difference exists between African and PNA countries, and this dimension is one that African countries can act on immediately. PNA countries sell access as a cartel, much like OPEC does for oil, plausibly garnering market power that enables them to extract greater profit from access agreements. In contrast, African countries sell access individually, so a buyer can negotiate for the lowest possible price, which substantially reduces the potential for rent extraction by African countries. However, African countries are becoming more economically integrated, as exemplified by the recently implemented African Continental Free Trade Area[10]. If African countries made the same choice as the PNA to sell their access rights as a bloc instead of as individual countries, how would profits, catch, and fish stocks in African waters change?

What complicates this story is that there is also market power on the demand side[11]. For example, since European countries *buy* access as a bloc, then the market power of the buyer competes against the market power of the seller. Indeed, most economic models of market power allow only for market power on the seller (supply) side or the buyer (demand) side. For example, monopoly models allow the actions of the monopolist to affect the equilibrium price (e.g., access fee), but they assume that the actions of buyers cannot affect the equilibrium price.

Here, we require a more flexible model. Since there are a small number of countries that buy and sell fishing access, we need a model in which the actions of every foreign buying country and every African selling country affect both the access fee and the allowed quantity of catch. To accommodate this reality, we use a "bilateral oligopoly" model[12]. We develop, parameterize, and estimate a bilateral oligopoly model that allows for market power on both the demand and supply side of the international fishing access market. We combine that model with a simple bioeconomic model, which allows us to calculate the rent capture and ecological consequences of an African fish cartel.

## Results

We compare two scenarios: the "status quo" scenario in which African countries sell access as individual countries, and the "coalition" scenario, in which African countries sell access as a bloc. This institutional difference is the only difference between the two scenarios; everything else is the same, such as enforcement against illegal fishing, legal fishing costs, and the degree of fish stock movement between national and international waters. We are therefore able to isolate the effects of enhanced African market power on profits, catch, and fish stocks, holding other institutions and parameters such as fish price fixed[13].

How would forming a fish cartel help Africa? Essentially, a cartel would allow African countries to exercise market power by restricting catch in African waters. Importantly, this cannot be accomplished in the absence of a cartel because when one country acts to restrict catch, the access fee rises, causing other countries to allow more catch in their waters. Instead, with a cartel, we estimate that the Africa Coalition would restrict catch allowed under access agreements by 29%,

increasing the equilibrium access fee by 19%. The African Coalition gives rise to both economic gains (to Africa) and increased fish abundance.

### Calculation of policy functions

There are important feedback dynamics to consider. The change in catch by foreign vessels under access agreements in African waters could affect the quantity of "non-access" catch in African waters. Non-access catch is any catch not covered by access agreements, such as catch by domestic fishers or unauthorized catch by foreign fishers. Since access catch decreases in the coalition scenario, and the fish stock therefore increases, non-access catch may increase in response because there are now more fish in the water that are available to catch. Accounting for this response is necessary for calculating how the Africa Coalition would affect overall biomass (tons of fish in the water). We do so by numerically deriving the access and non-access policy functions of each country from our bilateral oligopoly model. Given a fish stock biomass, a country's policy function returns the quantity of access catch and non-access catch, where it is intuitive that both policy functions are increasing in the total biomass of available fish.

To illustrate these findings, Fig. 1 displays the policy functions for a single African country (Madagascar), which has the median biomass among all African countries. As the biomass of fish in Madagascar's waters increases, the country rationally responds by allowing foreign fishers to catch more fish under access agreements. As noted above, non-access catch also increases with biomass, and at a faster rate than access catch. Most catch in African waters is non-access, illustrating the importance of explicitly modeling this component of the market. We calculate the total catch policy function as the access catch policy function plus the non-access catch policy function.

The Africa Coalition changes countries' access policy functions: at any given level of biomass, countries allow less access catch in their waters, which increases the access fee and profits they receive, as well as the biomass in their waters. We assume countries' non-access policy functions are the same in both scenarios; while increases in biomass are still met with an increase in non-access catch, the function itself is unchanged as a result of the coalition. In fact, it is plausible that at any given level of biomass, there would actually be less non-access catch in an Africa Coalition (in other words, the coalition could cause the non-access policy function to pivot down). For example, if the coalition makes countries more able to deter illegal fishing, then the gain in biomass would be even larger than our estimates. But if the opposite occurs, then our estimates would overstate the gain in biomass. For example, if domestic fishers' catch increases at any given biomass level due to reduced competition from foreign vessels operating under

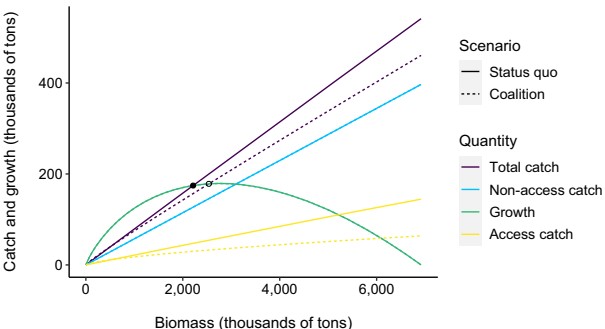

**Fig. 1 | Policy functions and growth curve for median biomass country (Madagascar).** The *x*-axis is biomass and the *y*-axis is catch or growth of the stock at that level of biomass. Points indicate equilibria in the status quo (solid) and Africa Coalition (hollow) scenarios. Under the Africa Coalition, all countries supply less access catch at any given level of biomass. This rightward rotation of countries' access catch and total catch policy functions increases equilibrium biomass.

access agreements, then the gain in biomass from the Africa Coalition would be smaller than our estimates.

## Obtaining equilibria

How do biology and policy functions interact? The total catch policy function returns the total annual catch a country would allow for each level of biomass in its waters. The biological growth curve, derived from country-specific fisheries data, returns the growth of the fish stock for a given level of biomass. If the growth is higher than the policy function-derived harvest (which occurs for relatively low levels of biomass), then the biomass increases. If the growth is lower than the harvest, then the biomass decreases. The equilibrium in each scenario occurs at a biomass level where the total catch policy function crosses the growth curve (indicated by the points in Fig. 1). Our analysis, therefore, compares the equilibrium under contemporary conditions to the new equilibrium that would occur (after an adjustment period) under an Africa Coalition.

## Continent-level results

We sum equilibria over all African countries to display continent-level results in Table 1. Because all countries allow less access catch at any given level of biomass, the continent-level equilibrium access catch decreases to 1.75 million tons per year, from 2.47 million tons in the status quo. The reduction in supply to the access market increases the equilibrium access fee from $128.20 in the status quo to $152.30 in the Africa Coalition scenario. African countries consequently increase their profit from $162 million per year to $199 million per year, while the profit of foreign buying countries decreases from $363 million to $306 million per year. The increase in non-access catch more than offsets the decrease in access catch (this is possible within ecological constraints due to the overall increase in biomass, which we address below). Non-access catch, which includes both domestic and unauthorized foreign catch, increases from 9.46 million tons to 10.36 million tons per year. Total catch in African waters is actually slightly higher with the Africa Coalition than under the status quo, but because countries allow less total catch at any given level of biomass, biomass is also higher for all countries (countries' total catch policy functions intersect their growth curves at higher biomass levels). We estimate that the Africa Coalition would increase total biomass in African waters from its current level of 117 million tons to 136 million tons, an increase of 16%. The 16% increase in biomass is why non-access catch can more than compensate for the reduction in access catch. These results indicate that an African Coalition would deliver both economic and ecological benefits for African countries, resulting in higher profits, higher total catch, and larger biomass. Our findings are robust to a range of alternative specifications and assumptions (Supplementary Figs. S1–S16 and Tables S1–S7).

## Country-specific results

While these results are quite positive at the continent-level, the Africa Coalition has specific implications for each African coastal nation. We display in Fig. 2 the percentage changes in access catch, profit, total catch, and biomass for each country. These percentage changes are relative to status quo values (Supplementary Fig. S17). Under the Africa Coalition, access catch declines in all countries (ranging from 24% to 36% decline) and profit increases in all countries (by 17% to 28%). Total catch slightly increases for all countries, as the increase in non-access catch more than compensates for the decrease in access catch (Supplementary Fig. S18). Biomass increases for all countries as well, from a minimum of 2.3% in Angola to a maximum of 31.6% in Cape Verde. On average, biomass and profit increase by 15% and 23%.

## Regional coalition scenario

Regional coalitions, in which African countries form cartels with their geographic neighbors, could serve as an intermediate stage between the status quo and a continent-level cartel. We calculate the benefits of regional coalitions by applying our bilateral oligopoly model to a counterfactual scenario based on Africa's eight existing Regional Economic Communities. We find that regional coalitions would increase biomass in African waters by 3 million tons (2%) and profits by $5 million per year (3%) compared to the status quo (Table S8 and Supplementary Figs. S19–S21). These gains are small because regional cartels only modestly increase African market power. While regional cartels may be more feasible in the short-term, the gains from a continent-level cartel are much larger.

## Discussion

We estimate that if Africa was to organize as a fishing cartel, the continent would stand to gain more profit (+23%), total harvest (+1.5%), and fish biomass (+16%) relative to the status quo. These gains occur because the Africa Coalition incentivizes conservation: countries earn more profit from restricting access catch, which increases the access fee as well as biomass. Creating and sustaining such an institution would require complex negotiations and regular cooperation among member states. But the fact that all African countries would benefit could make the creation of an Africa Coalition more feasible. Future research could quantify the costs associated with the formation and maintenance of an Africa Coalition, enabling a comparison with the benefits we calculate here.

Foreign fishers facing an Africa Coalition would secure a smaller share of the rents; we estimate that these foreign fleets would experience a drop in profit of 16%. Total economic surplus–seller profits plus buyer profits–is 4% lower under the Africa Coalition. But that measure of economic surplus does not include the non-market value of increased biomass in African waters, the economic value of increased non-access catch, the possible impact on economic growth if the increase in profit is invested rather than consumed, increased foreign exchange earnings, the potential for job creation in both harvesting and throughout the value chain, or the gain in equity from shifting profits from richer buying countries to poorer selling countries. For instance, 48% of non-access catch in African sellers' waters is domestic, rather than unauthorized foreign[14]. Increased domestic

## Table 1 | Effect of Africa Coalition on continent-level catch, profit, and biomass

|  | Status quo | Coalition | Difference | % Difference |
|---|---|---|---|---|
| Catch, access (millions of tons) | 2.47 | 1.75 | −0.72 | −29% |
| Access fee per ton | $128.20 | $152.30 | $24.10 | 19% |
| African sellers' profit (millions) | $162.15 | $199.24 | $37.09 | 23% |
| Foreign buyers' profit (millions) | $363.15 | $305.55 | −$57.60 | −16% |
| Catch, non-access (millions of tons) | 9.46 | 10.36 | 0.90 | 10% |
| Catch, total (millions of tons) | 11.93 | 12.11 | 0.18 | 2% |
| Biomass (millions of tons) | 117.20 | 136.12 | 18.93 | 16% |

Catch and profit are annual quantities. Access fee and profit are in 2020 USD.

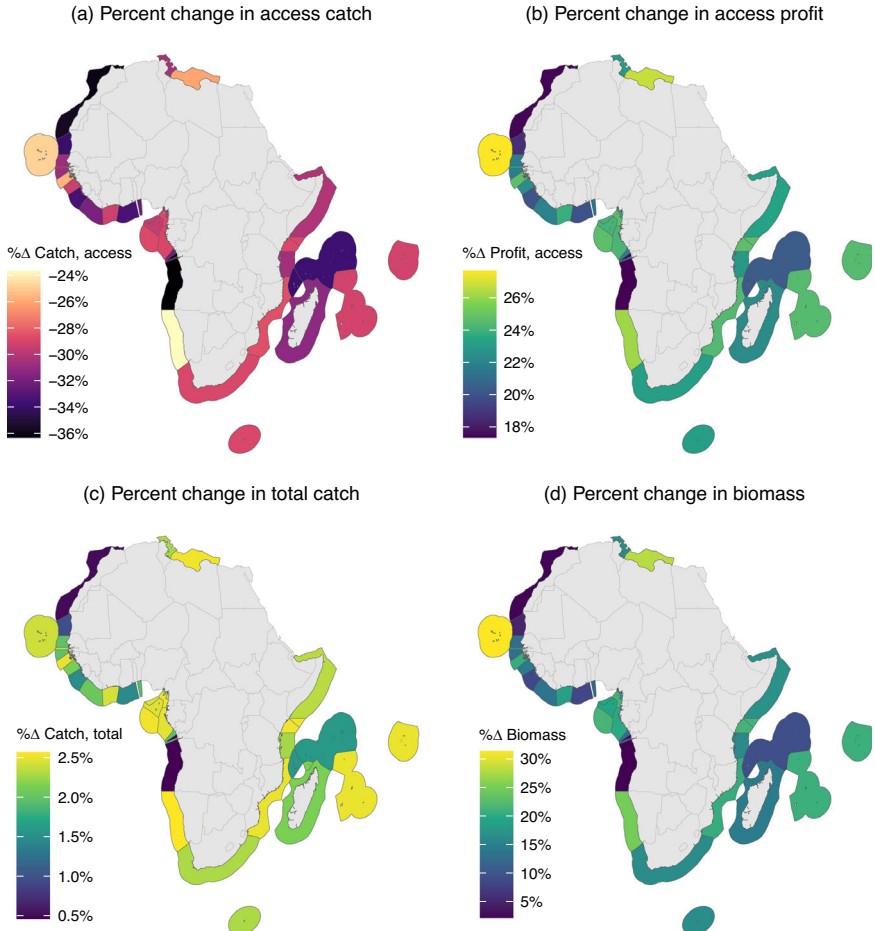

**Fig. 2 | Effect of Africa Coalition on. a** access catch, **b** profit, **c** total catch, and **d** biomass, by selling country. Percent changes are relative to status quo values. Total catch is access catch plus non-access catch. We define selling countries with a fishing hours-based threshold rule (see Methods).

catch could lead to larger fish consumption for coastal communities, and less competition for fishing grounds from foreign vessels operating under access agreements could result in higher profits for domestic fishers. While it is beyond the scope of this paper, valuing each of these other benefits would likely imply that the Africa Coalition raises global welfare (and, as we have shown, unambiguously raises welfare in Africa).

We have used our bilateral oligopoly model to estimate the consequences of an African Coalition, inspired by the successful PNA coalition among Pacific Island nations. We can also use our model in reverse to estimate the losses that would occur were the PNA to backslide into a situation where each country sells access individually (Tables S9–S11 and Supplementary Figs. S22–S27). In that case, the PNA coalition is the status quo scenario and the nine countries selling access individually is the counterfactual scenario. We find that if the PNA coalition did not exist, biomass in these countries' waters would decrease by 16 million tons (52%) and profit earned by these countries would decrease by $113 million per year (41%).

Thus, we estimate that the benefits to PNA countries of a coalition are even larger than the benefits to African countries. The reason is access catch comprises a much larger share of total catch in PNA country waters than in African waters (65% compared to 21%). This difference reverses the effects of a coalition on access catch, total catch, buyer profit, the access fee, and total economic surplus from the PNA market. Without the coalition, competition by each country to supply access to their waters leads to substantial overfishing. In equilibrium, access catch and total catch would decline by 22% and 33%

because biomass would be so much lower. The profit of the countries that purchase the right to fish in PNA waters would also decrease by 19% for this reason. Effectively, our model suggests a substantial conservation benefit of the PNA, which in turn confers a benefit even to the buying countries. Since supply (biomass) would be so much lower without the coalition, the access fee would actually increase by 4%, though profit to each country would decline substantially. Finally, in the PNA market, market power has the uncommon effect of increasing total economic surplus because of its large positive effect on biomass. If the PNA coalition did not exist, total economic surplus would be 27% lower. Like the African countries we studied in this paper, the nine PNA countries gain substantially from a coalition. The PNA coalition that already exists provides a starting point for the design of a similar Africa Coalition.

We model the Africa Coalition in a simple framework to illustrate the benefits of Pan-African cooperation. Other more complicated mechanisms by which an African fish cartel could sell fishing access, such as coordinated auctions, would also likely increase biomass in African waters and profits earned by African countries. Future research could also model individual fish stocks, rather than aggregating fish stocks to the country-level.

African countries are already integrating economically in order to exercise market power outside of fisheries. Successor agreements to the African Continental Free Trade Area may enable African countries to negotiate free trade agreements jointly, as countries in the European Union do. Just as countries in the European Union purchase fishing access as a bloc, African countries may one day sell fishing

access as a bloc. We find that doing so would rebalance African countries' economic relationships with other countries, enabling them to extract more profit from the international fishing access market, and to rebuild fish stocks that are depleted in part due to colonial legacies[15,16].

## Methods

### Bilateral oligopoly model

We apply the bilateral oligopoly model of Hendricks and McAfee (2010) to the international market for fishing access[12]. Each selling country $i$ is endowed with a true biomass, $b_i$, and each buying country $j$ is endowed with a true fishing capacity, $t_j$ (total gross tonnage of $j$'s distant-water fishing vessels). The model in Hendricks and McAfee (2010) is based on "reports" of supply, which we differentiate from true values with the ˆ symbol. In our setting, each selling country reports a biomass, $\hat{b}_i$, and each buying country reports a fishing capacity, $\hat{t}_j$. Countries report simultaneously to the market mechanism, which chooses the equilibrium access fee to equate supply and demand, and allocates the quantity of access permits across countries efficiently.

Countries know how their report affects the equilibrium access fee and quantity. Each seller knows the sum of other sellers' reports, but they do not know the true biomass of other sellers. The same is true for buying countries. As in Hendricks and McAfee (2010), the extent to which a country's report understates its true biomass or true fishing capacity reflects that country's market power. In other words, the model allows countries to exercise market power by under-reporting their true biomass or fishing capacity. This is the market power mechanism within the model, rather than a claim about the actual negotiation tactics preceding access agreements. For instance, in reality, knowledge over true biomass and fishing capacity varies by country.

Selling Country $i$'s opportunity cost (or shadow cost) of supplying access, $c(\cdot)$, captures the value of the future stock growth that $i$ foregoes in supplying access to foreign fishers. It is constant returns to scale in $b_i$, taking the form $b_i \times c(\frac{q_i}{b_i})$, where $q_i$ is the quantity of permits $i$ supplies and $c(\cdot)$ is convex and strictly increasing. The units of $b_i$, $\hat{b}_i$, and $q_i$ are metric tons of fish. Seller $i$'s marginal opportunity cost, or shadow cost, increases in quantity supplied and decreases in $b_i$. The market mechanism sets $i$'s quantity share equal to their share of total reported biomass: $q_i = \frac{\hat{b}_i}{\hat{B}} Q(\hat{B}, \hat{T})$, where $\hat{B} = \sum_i \hat{b}_i$, $Q = \sum_i q_i$, and $\hat{T} = \sum_j \hat{t}_j$. Seller $i$ reports $\hat{b}_i$ to maximize profit from supplying fishing access:

$$\max_{\hat{b}_i} \; p(\hat{B}, \hat{T}) \frac{\hat{b}_i}{\hat{B}} Q(\hat{B}, \hat{T}) - b_i c\left(\frac{\hat{b}_i Q(\hat{B}, \hat{T})}{\hat{B} b_i}\right) \quad (1)$$

The first term in Equation (1) is Seller $i$'s revenue and the second term is $i$'s opportunity cost of supplying access. Since Seller $i$'s report affects $\hat{B}$, Seller $i$ accounts for the fact that their report affects both $p$, the equilibrium access fee, and $Q$, the equilibrium total quantity of access permits (equivalently, the equilibrium total quantity of access catch in tons).

Buying country $j$ earns fishing profit with the function $t_j \times v(\frac{q_j}{t_j})$, where $q_j$ is the quantity of access permits purchased by $j$ and $v(\cdot)$ is concave and strictly increasing. The quantity of permits purchased by $j$ equals the quantity of fish caught by $j$. Fishing profit equals the revenue earned from fishing (ex-vessel price times quantity of catch), minus the fleet-level physical cost of fishing (e.g., fuel, labor, and capital costs). Fishing profit does not include the cost of the permits themselves, which are subtracted from $j$'s final profit separately. Buyer $j$'s marginal fishing profit decreases in the quantity of access permits because fishing costs increase in the quantity of fish caught. Buyer $j$'s marginal fishing profit increases in true fishing capacity because countries with greater fishing capacity can catch fish at a lower cost. The market mechanism sets $j$'s quantity share equal to their share of total reported gross tonnage: $q_j = \frac{\hat{t}_j}{\hat{T}} Q(\hat{B}, \hat{T})$. Buyer $j$ reports fishing capacity $\hat{t}_j$ to

maximize fishing profit minus permit cost:

$$\max_{\hat{t}_j} \; t_j v\left(\frac{\hat{t}_j Q(\hat{B}, \hat{T})}{\hat{T} t_j}\right) - p(\hat{B}, \hat{T}) \frac{\hat{t}_j}{\hat{T}} Q(\hat{B}, \hat{T}) \quad (2)$$

The first term in Equation (2) is Buyer $j$'s fishing profit and the second term is Buyer $j$'s cost of purchasing access permits. Since Buyer $j$'s report affects $\hat{T}$, $j$ accounts for the fact that their report affects both $p$ and $Q$.

Hendricks and McAfee (2010) obtain a closed-form solution when opportunity cost and fishing profit elasticities are constant; we adopt a similar approach. Let $c(z) = \frac{\eta}{\eta + 1} z^{(\eta + 1)/\eta}$ and $v(z) = \frac{\epsilon}{\epsilon - 1} z^{(\epsilon - 1)/\epsilon}$, where $\eta > 0$ and $\epsilon > 1$. Then given any vector of reports, the equilibrium price and quantity are

$$p(\hat{B}, \hat{T}) = \hat{B}^{-1/(\epsilon + \eta)} \hat{T}^{1/(\epsilon + \eta)} \quad (3)$$

and

$$Q(\hat{B}, \hat{T}) = \hat{B}^{\epsilon/(\epsilon + \eta)} \hat{T}^{\eta/(\epsilon + \eta)} \quad (4)$$

The equilibrium price decreases in reported supply ($\hat{B}$) and increases in reported demand ($\hat{T}$), while the equilibrium quantity increases in both reported supply and reported demand. The equilibrium price and quantity occur from equating the partial derivatives of opportunity cost and fishing profit (Supplementary derivations). The market mechanism therefore finds the price and quantity that maximizes total economic surplus assuming sellers and buyers' reports are truthful. The market mechanism is efficient conditional on reports, but when reports differ from true biomass or gross tonnage values, the resulting equilibrium will not maximize total economic surplus.

Equation 18 in Hendricks and McAfee (2010) expresses the relationship between reports and true values. Re-writing that equation in the language of our model, we have

$$\begin{aligned} \frac{\hat{b}_i}{b_i} &= \left(1 - \frac{\sigma_i}{\epsilon + \eta \cdot (1 - \sigma_i)}\right)^{\eta} \\ \frac{\hat{t}_j}{t_j} &= \left(1 + \frac{s_j}{\epsilon \cdot (1 - s_j) + \eta}\right)^{-\epsilon} \end{aligned} \quad (5)$$

where $\sigma_i = \frac{\hat{b}_i}{\hat{B}}$ and $s_j = \frac{\hat{t}_j}{\hat{T}}$. Writing Equation (5) in terms of quantity shares $\sigma_i$ and $s_i$ provides intuition regarding the market power mechanism in our model. The larger the share of the market a seller or buyer represents, the more they will understate their biomass or gross tonnage capacity.

We re-arrange Equation (5) to solve for Seller $i$'s equilibrium report given the reports of all other sellers, $\hat{b}_{-i} = \sum_{k \neq i} \hat{b}_k$, and $i$'s true biomass. We obtain

$$\epsilon \hat{b}_i^{(\eta + 1)/\eta} + b_i^{1/\eta}(1 - \epsilon)\hat{b}_i + \hat{b}_{-i}(\epsilon + \eta)\hat{b}_i^{1/\eta} - b_i^{1/\eta}\hat{b}_{-i}(\epsilon + \eta) = 0 \quad (6)$$

We solve for $\hat{b}_i$ numerically.

We also re-arrange Equation (5) to solve for Buyer $j$'s equilibrium report given the reports of all other buyers, $\hat{t}_{-j} = \sum_{k \neq j} \hat{t}_k$, and $j$'s true fishing capacity. We obtain

$$(\eta + 1)\hat{t}_j^{(\epsilon + 1)/\epsilon} - \eta t_j^{1/\epsilon}\hat{t}_j + \hat{t}_{-j}(\epsilon + \eta)\hat{t}_j^{1/\epsilon} - t_j^{1/\epsilon}\hat{t}_{-j}(\epsilon + \eta) = 0 \quad (7)$$

We solve for $\hat{t}_j$ numerically. We use Equations (3)–(7) to solve for the Nash Equilibrium with a fixed point algorithm. The Nash Equilibrium is the set of reports where no seller or buyer could increase their profit by changing their own report, given the reports of other sellers and buyer.

Theorem 7 in Hendricks and McAfee (2010) states (in the language of our model) that as the number of selling countries decreases, holding total true biomass constant, (1) total reported biomass decreases, (2) the equilibrium access fee increases (because reported

supply decreases while demand remains unchanged), and (3) the equilibrium total quantity of access permits decreases. Additionally, profit always increases for the countries that join the coalition, but it does not necessarily increase for countries outside the coalition. Our results accord with this theorem.

**Setting up the model**
We define the market as the universe of countries that sell and buy access to African waters. Then, we calculate the two parameters necessary for solving our bilateral oligopoly model: each selling country's true biomass $b_i$ and each buying country's true gross tonnage $t_j$.

**Identifying sellers and buyers.** We use apparent fishing effort data from Global Fishing Watch (GFW) to identify sellers and buyers[17]. GFW uses machine learning algorithms to predict fishing activity from Automatic Identification System (AIS) vessel movements. The data provides predicted hours of fishing at the vessel-day-.1° grid cell level. The names of these data files on GFW's website are mmsi-daily-csvs-10-v2-[year].zip. We filter the data to grid cells that occur inside the Exclusive Economic Zone (EEZ) of an African country[18]. GFW's data begins in 2012, but we use fishing data beginning in 2016 because that is the year in which satellite coverage to receive AIS transmissions significantly improved.

For each year between 2016 and 2020, we calculate the pairwise buyer-seller total fishing hours. For example, one observation in our pairwise data is the fishing hours of all Chinese-flagged vessels inside Senegal's EEZ in 2016, a second observation is the fishing hours of all European Union-flagged vessels inside Senegal's EEZ in 2019, and a third observation is the fishing hours of all Chinese-flagged vessels inside Kenya's EEZ in 2018. Given some threshold of fishing hours, we conclude that a country is a buyer in the "African Access Market" if in at least 1 year it fishes for more hours than the threshold in at least one African country. We similarly conclude that an African country is a seller if, in at least 1 year vessels flagged to a foreign country fish for more hours than the threshold.

Supplementary Fig. S1 displays the number of sellers and buyers that we would include as participants in the African Access Market as the fishing hours threshold varies. We collect data on all publicly available access agreements between 2010 and 2021. We set the fishing hours threshold as the maximum value that would include all selling countries for whom we observe an access agreement between 2016 and 2020 as participants in the African Access Market. In other words, we choose the largest fishing hours threshold that would not exclude any selling country that we know sells access to its waters. This rule yields 32 selling countries. We use the same threshold to identify 33 buying countries; in at least one year, all of these countries fish at least as many hours as the threshold in at least one selling country. The threshold is 2,420 hours.

Our results are robust to misclassification of buyers because we hold the demand-side fixed in our coalition simulation. However, including countries as sellers who do not actually participate in the African Access Market will cause us to overestimate the gains to African countries of the coalition. We assess the robustness of our results to this potential misclassification by doubling our preferred fishing hours threshold and repeating our analysis (Table S1 and Supplementary Figs. S3, S4). Our results are almost identical. We estimate that a coalition would increase total biomass in African waters by 18 million tons and total seller profits by $35 million (compared to 19 million tons and $37 million in our preferred specification).

**Calculating each buying country's true gross tonnage.** For each vessel flagged to a buying country, we calculate the fishing hours between 2016 and 2020 that occur inside the buying country's EEZ (domestic fishing) and the fishing hours that occur outside the buying

country's EEZ (distant-water fishing). We classify vessels as distant-water fishing (DWF) vessels if more than half of their fishing hours occur outside the EEZ of the country they are flagged to. We obtain an estimate of the gross tonnage of each of these DWF vessels from a second GFW dataset. This dataset is called fishing-vessels-v2.csv on GFW's website. We sum gross tonnage over DWF vessels to obtain the total gross tonnage of each buying country's DWF fleet (Supplementary Fig. S2).

**Calculating each selling country's true biomass.** First, we suppose that for every selling country, true biomass in the status quo $b$ divided by biomass at maximum sustainable yield $b_{MSY}$ is 0.8. (We omit Selling Country $i$ subscripts in this subsection.) The value of 0.8 reflects stocks that are partially depleted. Biomass estimates are largely unavailable for African stocks[19]. We verify the robustness of our results to this assumption in Tables S2, S3 and Supplementary Figs. S5–S8. In the Pella-Tomlinson surplus production model,

$$b_{MSY} = \frac{k}{(\phi + 1)^{1/\phi}} \tag{8}$$

where $k$ is the country's carrying capacity and $\phi$ is the shape (of the growth curve) parameter[20]. We set $\phi = 0.188$ for all countries so that $b_{msy}$ occurs at 40% of carrying capacity[21,22]. We substitute $\frac{b}{b_{MSY}} = 0.8$ into Equation (8) to obtain $b = \frac{.8k}{(\phi+1)^{1/\phi}}$. We then substitute this expression for $b$ into the Pella-Tomlinson equilibrium condition to obtain $k$ as a function of catch (or harvest) $h$, growth parameter $g$, and $\phi$:

$$
\begin{aligned}
h &= \frac{\phi+1}{\phi} g b \cdot \left(1 - \left(\frac{b}{k}\right)^\phi\right) \\
&= \frac{\phi+1}{\phi} g \frac{.8k}{(\phi+1)^{1/\phi}} \cdot \left(1 - \left(\frac{\frac{.8k}{(\phi+1)^{1/\phi}}}{k}\right)^\phi\right) \\
&= .8gk \frac{(\phi+1)^{\frac{\phi-1}{\phi}}}{\phi} \cdot \left(1 - \frac{.8^\phi}{(\phi+1)^{\phi/\phi}}\right) \\
&= .8gk \frac{(\phi+1)^{\frac{\phi-1}{\phi}}}{\phi} \cdot \frac{\phi+1-.8^\phi}{\phi+1} \Rightarrow
\end{aligned}
\tag{9}
$$

$$
\begin{aligned}
k &= h \frac{1.25}{g} \cdot \frac{\phi}{(\phi+1)^{\frac{\phi-1}{\phi}}} \cdot \frac{\phi+1}{\phi+1-.8^\phi} \\
&= h \frac{1.25}{g} \phi \frac{(\phi+1)^{\frac{1}{\phi}}}{\phi+1-.8^\phi}
\end{aligned}
\tag{10}
$$

We measure $h$ as total average catch in the country's waters between 2010 and 2018 with estimates from the Sea Around Us[14]. 2018 is the most recent year in the Sea Around Us data. Total catch includes foreign catch allowed under access agreements, domestic catch, discards, and unauthorized catch by both domestic and foreign fishers. Our estimate of $g$ is the weighted average $g$ over all stocks in each country from ref. 23. We weight each country's $g$ by the carrying capacity of each stock inside the country's waters, using estimates of carrying capacity from Costello et al. (2016). We only use Costello et al. (2016) carrying capacity estimates as weights in calculating average $g$. We calculate our own value of carrying capacity $k$ for each country in Equation (10). Given $k$, we also obtain $b$ since $b = \frac{.8k}{(\phi+1)^{1/\phi}}$. Supplementary Fig. S2b displays $b$, each selling country's biomass.

Our results are robust to the assumption that $\frac{b}{b_{MSY}} = 0.8$. When we instead assume $\frac{b}{b_{MSY}} = 0.6$ and repeat our analysis, we estimate that a coalition would increase total biomass in African waters by 21 million tons and total seller profits by $40 million (compared to 19 million tons and $37 million in our preferred specification). We display results for our $\frac{b}{b_{MSY}} = 0.6$ specification in Table S2 and Supplementary Figs. S5, S6.

As an additional robustness check, we repeat our analysis assuming $\frac{b}{b_{MSY}} = 0.4$. In this case, we estimate that a coalition would increase total biomass in African waters by 25 million tons and total seller profits by $48 million (Table S3 and Supplementary Figs. S7, S8).

## Estimating the model

We use our bilateral oligopoly model to simulate the change in the equilibrium access fee, catch, and profits when African countries sell access as a bloc instead of as individual countries. We refer to this counterfactual scenario as the "coalition" scenario, and the scenario with individual countries selling access as the "status quo". The only difference between the two scenarios is there is one selling country in the coalition scenario and 32 selling countries in the status quo scenario.

Each selling country has its own fish stock in both scenarios: stocks stay within their original, status quo boundaries in the coalition scenario. Moreover, the growth curve for each country's stock remains unchanged in the coalition scenario. However, the size of each country's stock increases in the coalition scenario for the following reason. In the status quo, a single country restricting supply (reported biomass) has little effect on the equilibrium access fee because other countries respond to any increase in the fee by increasing supply. By contrast, the existence of a single "country" in the coalition scenario eliminates the opportunity for other countries to offer compensating increases in supply when the access fee increases. Shutting down this compensating mechanism enables effective restriction of total access catch, which increases biomass in the waters of every country (by different amounts), as well as the access fee.

Following Hendricks and McAfee (2010), to make units comparable, we convert the true gross tonnage of buying countries and the true biomass of selling countries into gross tonnage shares and biomass shares. Countries report their gross tonnage share or biomass share in our simulations. The sum of the gross tonnage shares over countries equals 1 in both scenarios, as does the sum of the biomass shares in the status quo scenario. In the coalition scenario, the single "country" has a true biomass share of 1.16. This value equals the true equilibrium biomass in the coalition scenario divided by the true total (summed over countries) biomass in the status quo scenario. In addition to the benefit of making units comparable, normalizing gross tonnage and biomass as shares reduces the influence of measurement error on our results. For example, if measurement error is constant across countries (e.g., we incorrectly specify 10% of each countries' gross tonnage as distant-water instead of as domestic), then errors in measuring gross tonnage and biomass would not affect our results.

We find that the equilibrium access fee is 19% higher and the equilibrium access quantity is 29% lower in the coalition scenario compared to the status quo scenario. These differences occur because the Africa Coalition reports 40% lower biomass than the total biomass reports of the 32 individual African countries, even though total biomass in the Africa Coalition is 16% higher.

Our bilateral oligopoly model illustrates how an Africa Coalition might change the quantity of catch allowed under access agreements, but by itself, it does not reveal how total catch (access catch plus non-access catch) and biomass would change. We, therefore, numerically derive access and non-access policy functions for each country. Given a value of biomass, a country's policy function returns the quantity of access catch or non-access catch. The total catch policy function equals the sum of the access and non-access policy functions.

We set $\eta = 1$ and $\epsilon = 2$ in both scenarios. Recall that $\eta$ must be greater than 0 and $\epsilon$ must be greater than 1. We set $\eta = 1$ and $\epsilon = 2$ because these values are both 1 unit above their minimum allowed values. We assess the robustness of our results to these parameter value choices by repeating our analysis with $\eta = 0.5$, $\eta = 1.5$, $\epsilon = 1.5$, and $\epsilon = 2.5$ (Tables S4–S7 and Supplementary Figs. S9–S16). Across these four robustness checks, we estimate that a coalition would increase total biomass in African waters by between 12 and 26 millions tons, and total seller profits by between $25 and $58 million.

**Status quo access policy functions.** First, we numerically derive the status quo access policy function of each country. The solid yellow line in Fig. 1 displays Madagascar's status quo access policy function as an example. Holding everything else fixed, we replace Madagascar's true biomass with a different value (x-axis), solve for the new Nash Equilibrium of our bilateral oligopoly model, and calculate Madagascar's resulting equilibrium access quantity (y-axis). We repeat this procedure from an initial biomass of near-zero up to Madagascar's carrying capacity, in increments of 1% of carrying capacity.

Before running the bilateral oligopoly model, we scale Madagascar's new biomass by its true biomass share. For example, Madagascar's true biomass is 2,213,411 tons and its true biomass share is .0189. When we run the bilateral oligopoly model with a new biomass of 3,458,613 tons, the true biomass share we input into the model is .0295 (because .0189 · (3,458,613/2,213,411) = 0.0295). When the new biomass is below true biomass, true total biomass in the model is slightly below 1, and vice versa; in this example, the true total biomass in the model is 1.0106.

After running the model, we scale the equilibrium total access quantity with data we collected on European Union (EU) access agreements between 2010 and 2021. Unlike other buying countries, the EU publishes all of its access agreements. In the status quo scenario, the equilibrium total access quantity in quantity model units is 0.935, of which the EU purchases 11.5%. According to access agreements, the EU purchased the right to catch 283,640 tons in African waters each year on average. We use these two EU values to calculate that the equilibrium total access quantity in tons is 2.47 million per year (283,640/0.115 = 2.47 million). When we run alternative versions of our bilateral oligopoly model, such as when we change the biomass of one selling country or when we run the coalition scenario, we convert the equilibrium total access quantity from quantity model units to tons using this relationship that 0.935 quantity model units equals 2.47 millions tons. In the above example where we set Madagascar's biomass as 3,458,613 tons but changed nothing else, the equilibrium access quantity in quantity model units is 0.942, which we calculate equals 2.49 million tons.

By changing one country's biomass at a time, re-running the bilateral oligopoly model, scaling total equilibrium access quantity into tons, and apportioning access quantity to each country, we construct the access policy function for every country. Recall that in the model each country's access quantity equals the equilibrium total access quantity times the country's share of reported biomass. We find that access policy functions are close to linear (Fig. 1 is a representative example).

**Status quo non-access and total catch policy functions.** Given our finding that status quo access policy functions are approximately linear in biomass, we suppose that status quo non-access policy functions are linear as well. We calculate the slope of this function for Selling Country $i$ as $(h_i - q_i^*)/b_i$, where all values are in tons, $h_i$ is total catch according to Sea Around Us, $q_i^*$ is the equilibrium access quantity caught in country $i$'s waters according to our bilateral oligopoly model, and $b_i$ is true biomass. We can then calculate non-access catch for every country at any level of biomass. The solid blue line in Fig. 1 displays Madagascar's non-access policy function. We sum the access and non-access policy functions of each country to obtain each country's total catch policy function.

**Coalition access, non-access, and total catch policy functions.** We calculate the coalition access policy function in a similar manner as the status quo access policy functions. Holding everything else fixed, we replace true biomass with a different value, solve for the new Nash

Equilibrium of our bilateral oligopoly model, and calculate total equilibrium access quantity. We repeat this procedure from an initial biomass of near-zero up to the total carrying capacity of all 32 selling countries, in increments of 1% of total carrying capacity. We scale total equilibrium access quantity into tons in the same manner as in the status quo scenario. One difference in the coalition scenario is we cannot apportion access quantity to countries according to their biomass reports, because there is only one "country" in the coalition scenario. We, therefore, assign access quantities to countries in proportion to countries' true status quo biomass. For example, since Madagascar holds 1.89% of status quo biomass, we assign Madagascar 1.89% of access quantity in the coalition scenario. This apportioning allows us to construct access policy functions for each country in the coalition scenario. The dashed yellow line in Fig. 1 is Madagascar's coalition access policy function.

We assume countries' non-access policy functions are the same in the coalition scenario as they are in the status quo. We sum the access and non-access policy functions to obtain the coalition total catch policy function for each country. If countries' non-access policy functions rotated to the right like their access policy functions do, which could occur if the coalition makes countries more able to deter illegal fishing or more interested in limiting domestic fishing, then this assumption would result in our analysis underestimating the gain in biomass from the Africa Coalition. If instead countries' non-access policy functions rotate left, then this assumption would result in our analysis overestimating the increase in biomass from an Africa Coalition.

**Equilibrium counterfactual biomass.** Equilibrium counterfactual biomass for a country occurs at the intersection between the country's total catch policy function and its growth curve. Growth (measured in tons) for a country equals $\frac{\phi+1}{\phi}gb \cdot (1 - (\frac{b}{k})^{\phi})$, where $\phi = 0.188$, $g$ is the growth parameter, and $b$ is biomass (see discussion preceding Equation (9)). The coalition total catch policy functions we have constructed allow us to capture the response of non-access catch to the Africa Coalition. Though access catch decreases in the coalition scenario, non-access catch increases by a more than compensating amount, such that total catch with the Africa Coalition slightly exceeds total catch under the status quo (Table 1).

We find that equilibrium total biomass (summed over countries) is 16% higher in the coalition compared to the status quo. We use our policy functions to calculate access catch, non-access catch, and total catch at this level of biomass.

**Equilibrium access fee, seller profits, and buyer profits.** We calculate that the average per ton access fee is $128.20 (2020 USD). This value is the weighted average per ton access fee across all EU access agreements between 2010 and 2021, where the weights are the catch in tons allowed under each agreement. We do not include lump sum payments in this calculation. The status quo equilibrium access fee in fee model units is 0.961. When we run the coalition scenario, we convert the equilibrium access fee from fee model units to 2020 USD using this relationship that 0.961 fee model units equals $128.20. In the coalition scenario we find that the equilibrium access fee is 1.14 fee model units, which translates to $152.30.

We calculate seller and buyer profits using Equations (1) and (2). In the model, profit is denominated in fee model units times quantity model units. We convert profit into 2020 USD by multiplying profit by $128.20/0.961 (to convert from fee model units) and by 283,640/0.115 (to convert from quantity model units).

**Regional coalition scenario.** We use our bilateral oligopoly model to simulate a second counterfactual scenario. It may be more feasible in the short-term for African countries to form regional cartels, upon which a continent-level cartel could ultimately be created.

The African Union recognizes eight Regional Economic Communities (RECs)[24]. We group African selling countries into regional coalitions based on these RECs. Since RECs overlap, we use the following iterative decision rule to map each selling country to a single REC: start with the REC with the fewest selling country members and assign all of its members to this REC. For the next smallest REC, assign all of its members to this REC, except those countries that have already been assigned to a different REC. Continue applying this decision rule in ascending REC order. We make two adjustments to this algorithm. First, we exclude the Intergovernmental Authority on Development REC because it has only two selling country members. Second, we reassign the Democratic Republic of the Congo (DRC) from the East African Community (EAC) to the Economic Community of Central African States so that our regional coalitions are geographically continuous. Western Sahara is not a member of any REC, so we do not include it in any regional coalition; it participates in the access market as an individual country.

After creating our regional coalitions, we sum the biomass of countries in the same regional coalition and then we re-estimate our bilateral oligopoly model. This regional coalition scenario is identical to the continent-level scenario except there are now eight sellers reporting biomass instead of one (Supplementary Fig. S19).

We calculate access policy functions by replacing the true biomass of a region with a different value, solving for the new Nash Equilibrium, and calculating total equilibrium access quantity. We repeat this procedure one region at a time, holding everything else fixed, from an initial biomass of near-zero to the total regional biomass, in increments of 1% of regional biomass. As in the continent-level coalition scenario, we assign access quantities to countries in proportion to the share of true status quo biomass that they make up in their region. For example, Ghana comprises 7% of status quo biomass in ECOWAS (Economic Community of West African States) waters, so we assign 7% of ECOWAS's access catch to Ghana.

We assume non-access policy functions are the same as in the status quo, and we calculate total catch policy functions as the sum of each country's access and non-access policy functions. We identify each country's equilibrium biomass as the intersection of its total catch policy function with its growth curve. Finally, we use each country's policy functions to calculate access catch, non-access catch, and total catch at its equilibrium level of biomass. We conclude the scenario by scaling the equilibrium access fee and calculating buyer and seller profits with the same ratios we use in the continent-level coalition scenario.

African countries experience small benefits in this scenario because the regional coalitions do not sufficiently aggregate biomass (market power). ECOWAS, the largest regional coalition, holds 44% of Africa's total status quo biomass, but most regional coalitions hold much less; half of the regional coalitions each hold less than 4% of the continent's biomass. For this reason, biomass in ECOWAS country waters increases by 3% to 11% across countries, but countries that belong to regional coalitions with a smaller share of total biomass experience much smaller changes in biomass (Supplementary Fig. S21d). Profit increases by 2% to 7% across countries because all countries benefit from the slightly higher equilibrium access fee (Supplementary Fig. S20d and Table S8).

As in the status quo, most African countries are unable to meaningfully exercise market power in the regional coalition scenario. The sum of regions' equilibrium biomass reports is 89% of total true biomass. By contrast, in the continent-level coalition scenario the equilibrium biomass report is 50% of total true biomass. Recall that in our model sellers exercise market power by underreporting biomass, which increases the equilibrium access fee.

**Estimating the model for the PNA market.** We apply the same bilateral oligopoly model to the nine Pacific Island countries that comprise

the Parties to the Nauru Agreement (PNA). In the PNA market, the status quo scenario is a coalition and the counterfactual scenario is the nine countries selling access individually. We follow the same procedure as for the African market: identifying buyers with the same decision rule and fishing hours threshold, calculating the true gross tonnage of each buying country's distant-water fishing fleet, calculating each selling country's true total biomass with the assumption that $\frac{b}{b_{MSY}} = 0.8$, setting $\eta = 1$ and $\epsilon = 2$, deriving status quo and counterfactual policy functions, calculating equilibrium counterfactual biomass and the access fee, and calculating seller profits and buyer profits in both scenarios. Table S9 and Supplementary Figs. S22, S23 display the results. We identify 15 buying countries, compared to the 33 countries that buy access to African waters. We do not identify sellers with the fishing hours threshold rule because the identities of the nine selling countries are public knowledge.

In our main text discussion, we noted that buying countries benefit from the PNA in the sense that their profit would be lower if the PNA did not exist. This result raises the question of whether buying countries explicitly supported the creation of the PNA. While we could not find documentary evidence, our sense is that buying countries such as China, Japan, and Taiwan display partial support for the PNA's existence. In the short-run, buying countries may see the PNA as rent-decreasing, owing to the market power it allows selling countries to wield in the pricing of fishing access. With the long-run in mind, however, the PNA is regarded as having been instrumental in helping to ensure sustainable fishery management for most tuna stocks in the Pacific.

We calculate that the average annual access catch in PNA waters between 2010 and 2021 is 1.19 million tons[25]. We obtained this value by subtracting catch by the national fleet in national waters from the total catch in national waters. We assume all foreign catch in PNA waters is access catch because access catch data do not exist; there are only data on the number of days of access fishing. To assess robustness to this assumption, we instead assume half of foreign catch is access catch and then repeat our analysis (Table S10 and Supplementary Figs. S24, S25). We find that if the PNA did not exist, biomass in these countries' waters would decrease by 7 million tons and profit earned by these countries would decrease by $33 million (compared to 16 million tons and $113 million for our main specification). Therefore, the greater the fraction of foreign catch that is unauthorized, the more likely we are to overestimate the benefits of the PNA coalition. The fraction of foreign catch that is unauthorized appears to be quite low[26].

In our main specification when we assume access catch equals foreign catch, the equilibrium access catch supplied by the Solomon Islands in the status quo coalition scenario exceeds the Sea Around Us average annual total catch in the Solomon Islands' waters (172 thousand tons compared to 155 thousand tons). This difference results in a negative slope for the Solomon Islands' non-access policy function. Since negative non-access catch is impossible, we replace this slope with the average non-access policy function slope among the eight other PNA countries. Other than this adjustment, our analysis of the PNA market follows the same procedure as for the African market. The large effects for the Solomon Islands in Supplementary Figs. S22 and S23 occur because its status quo total catch policy function is steep. The robustness check when we assume access catch is half of foreign catch does not require us to adjust the slope of the Solomon Islands' non-access policy function because in this case status quo access catch is less than total catch in the Solomon Islands' waters.

PNA countries sell access in terms of fishing days, rather than tons. We calculate the average per-ton PNA access fee by dividing each country's license and access fee revenue by the quantity of foreign catch in each country's waters[25,27]. We calculate the average access fee between 2010 and 2019 because 2019 is the most recent year for which license and access fee revenue data are available. We calculate the

weighted average access fee as $307 (USD 2020), where the weights are the quantity of foreign catch in each country's waters each year.

Our final robustness check relates to our assumption of $\frac{b}{b_{MSY}} = 0.8$. This assumption enables a direct comparison of our primary PNA results to our primary Africa Coalition results. However, tuna stocks in the Western and Central Pacific Ocean—where PNA countries are located—are generally healthy, with $\frac{b}{b_{MSY}}$ values ranging from 0.42 to over 2.0 as per a 2017 assessment[28]. Examining the most recent stock assessments, which are complied in the Ram Legacy database[29], we find that these stocks continue to be healthy. For the purposes of this ancillary analysis, we use a value of $\frac{b}{b_{MSY}} = 1.3$ to illustrate the effects of dissolving the PNA in a setting where the stocks are reasonably healthy (Table S11 and Supplementary Figs. S26, S27). In this case, we find that biomass would decrease by 16 million tons and seller profit would decrease by $78 million (compared to 16 million tons and $113 million for our primary specification). The main differences from our primary specification is access catch would be 0.07 million tons higher and buyer profit would be $7 million lower (compared to 0.27 million tons lower and $89 million lower in our primary specification).

## Reporting summary
Further information on research design is available in the Nature Portfolio Reporting Summary linked to this article.

## Data availability
All data used in this study have been deposited in figshare at https://doi.org/10.6084/m9.figshare.23948991.

## Code availability
All replication code is available in figshare at https://doi.org/10.6084/m9.figshare.23948991.

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

## Acknowledgements
Yutian Fang and Halley McVeigh provided excellent research assistance. Abdoulaye Cisse, Felipe Jordán, Juan Carlos Villaseñor-Derbez, Megan Lang, Patrick Behrer, and Dale Squires contributed helpful comments on the manuscript. Erin O'Reilly supplied able project management. G.E. and C.C. benefited from generous funding for this work from The Nature Conservancy. The findings, interpretations, and conclusions expressed in this paper are entirely those of the authors. They do not necessarily represent the views of the World Bank and its affiliated organizations, or those of the Executive Directors of the World Bank or the governments they represent.

## Author contributions
G.E.: contributed conceptualization, methodology, investigation, visualization, project administration, supervision, and writing. C.C.: contributed conceptualization, methodology, funding acquisition, project administration, supervision, and writing.

## Competing interests
The authors declare no competing interests.
