## [Peer Review File · Nature Communications]

Reviewers' Comments:

Reviewer #1:

Remarks to the Author:

This is an interesting paper addressing an important issue in terms of natural resource extraction in developing countries. The paper is generally well motivated and executed. However, there is one part of the empirical outcome that warrants more discussion. The result that by reducing the access of foreign fishers will lead to both higher total landings and larger fish stocks seems counter-intuitive and almost too good to be true as most African nations do not manage their fisheries very effectively.

Ln 3. I do not like the start of the abstract. It assumes that the reader knows a lot about the trade with fishing rights. It would be good if this could be reformulated to make it more motivational.

Ln 18. While former colonizers are important in this trade, I think you should be explicit that there are also other agents with China probably being the most important today, and countries like South Korea and Taiwan are important. Until its collapse, also the USSR and other eastern block countries were important distant water fishing nations that bought or bartered fishing rights.

Ln 27. It would be good if you mentioned a few examples here before you focus on fish.

Ln 48. Would it not also make sense to put in management here? As I understand it, the PNA countries have a management system that is doing better in protecting stocks and thereby presumably increasing CPUE.

Ln 73. Fang and Asche (2021, Ecological Economics) provides evidence that the U.S. have buyer power for several species.

Ln 91. I think you should provide some more intuition for the outcomes here even though some of it is explained in more detail in the next section. I think it is particularly important to note that if the cartel is to exploit market power, the supply will be reduced and therefore the stock status improved (if the foreign catches are not completely crowded out by higher domestic landings), and as such that there is also likely to be a conservation benefit associated with the cartel.

You do not mention prices here. I think you should explicitly note that you regard the price of the fish as exogenous as there is a global market for most species (see e.g. Anderson et al., 2018, Journal of Commodity Markets).

Ln 145. I am a bit surprised at this result that total domestic landings increase more than the reduction in foreign landings. Can you please provide more intuition to explain this result as that would lead to a lower CPUE for domestic fishers and presumably lower profits.

Ln 148. And I really struggle to understand the result that fish stocks increase even with higher landings. That would be true if African fisheries are on the forward part of the backward-bending supply schedule, but that seems odd given the lack of management. I assume this follows from you partially depleted assumption (Ln 342), but is that not overly optimistic? Is 0.6 sufficient as a robustness check? For me it seems like that is also too optimistic as it still appears as you are not on the backward bending part of the supply schedule.

Reviewer #2:

Remarks to the Author:

My comments are attached below as a separate PDF document.

Review of "A fish cartel for Africa" (Gabriel Englander and Christopher Costello)
written by Philippe Marcoul

Background and policy questions: The economic problem analyzed in this work revolves around the sale of fishing access rights to their waters by African countries. Currently, all African countries *individually* negotiate fishing access with foreign countries (or a block of foreign countries), such as the European Union. These agreements usually specify an allowable tonnage for different species and an access fee per ton of fish caught in the Exclusive Economic Zone of the signing country. Moreover, it is notorious that these agreements are quite *unfavorable* to African countries, leaving them only with a tiny fraction of the value of the fish caught.

The paper's main question is simple: What would happen if African countries were cooperating and selling access rights as a cartel rather than separately?

The analysis conducted shows that if such a cartel was formed, African countries would widely benefit not only through a steep increase of profit earned (from the sale of access rights), but also, through a higher biomass in their waters.

In a related question, they also analyze the existing fish cartel formed by the Pacific Island countries (PNA). Taking an opposite view, they ask what would happen if the PNA cartel was dismantled and reverted to a situation where each Pacific island country was individually negotiating access rights with buying countries. The answer to this question shows that PNA countries have greatly benefited from the cartel formation.

Evaluation: I found this paper and its results highly relevant for three reasons. First, the idea and the contribution of the paper are novel and original. I am not aware of any comparable studies in the renewable resource literature. Second, the quantified evaluation that the authors can provide aligns with my economic expectations; I found these findings to be credible. My impression of credibility is reinforced by the methods used, which are both rigorous and well-executed. Last but not least, the subject of the study is highly topical and has far-reaching consequences. It will be of interest not just to resource and development economists but also to development practitioners and African fishery decision-makers of coastal countries. With

the quantified evaluations contained in the paper, the latter could find a solid starting point to engage in a conversation with each other.

Below, I develop a few comments/points that the authors may need to consider in revising the paper.

- The idea that non-access catch increases as a result of the coalition can, perhaps non-intuitively, be a good consequence of the coalition. This will be especially true if non-access catches are mostly made by subsistence fishers (or at least relatively poor ones). It would appear to me that if there is less access catch for foreign fishers, then local artisan fishers would be better off as a result of less competition on the fishing grounds. Hence, African countries would receive a benefit in the form of larger fish consumption for coastal communities (hardly a bad thing), and not necessarily in the form of a larger resource for the central state. If a poor state is subject to a high degree of resource misappropriation (e.g., corruption), then the former might be preferable to the latter. The authors should acknowledge this aspect, and perhaps develop it. Note that if the authors are to develop the discussion in this direction, then, it would be useful to report the change in non-access catch in Figure 2 of page 9.
- Related to the previous point. I am concerned by the assumption made on page 6, where it is assumed that countries' non-access policy functions "are the same in both scenarios." By switching to a cartel, it seems intuitive that smallholder fishers would compete (much) less harshly with foreign (international) fishers, who are usually better equipped. As such, the non-access catch policy function would change (not in a linear fashion). If the shape of the non-access catch policy function is different after the coalition is formed, then it is less clear how the benefits received by the African countries may change.
- Regarding the Africa Versus PNA coalition. An important observation is that the PNA, as analyzed by the authors, seems to be a Pareto optimum (see p.11). The PNA is thus stable in the sense that no player within this agreement has any incentives to break it. For instance, it would be counterproductive for buying countries to attempt to weaken the bargaining coalition formed by the (selling) Pacific Island Nations. In fact, buying countries may even subsidize the administrative costs of

organizing the coalition of selling countries. Is there any evidence/hint that buying countries have indeed favored the formation of the PNA cartel? If yes, this information should be reported as corroboration (or weak evidence) that the approach and calculations of the authors are correct.

In the case of Africa, this particular configuration is not present and sellers and buyers have opposite interests (even if total surplus is strictly larger). Buying countries may thus insist on dealing with an individual selling country.

- Report of fishing capacity (p. 14). How is the gross tonnage of country j , gt_j , determined? Does the auctioneer/market mechanism equate the following equation for country j

$$\frac{gt_j}{\sum_{j' \in J} gt_{j'}} = \frac{\widehat{gt}_j}{\sum_{j' \in J} \widehat{gt}_{j'}}?$$

I assumed it was a similar process to the one used for determining the quantity share q_i of the selling countries. Unless I missed it, I think this equation should be mentioned in the text.

- Determination of the equilibrium price (p.13-15). I did not find the determination of equations (3) and (4) easy. My understanding is that to do so, one must compute the supply functions of the selling countries and the demand for access permits of the buying countries, but also, one must use the two equations of the market mechanism (i.e., those linking reports with biomass and permit allocations). I would suggest that you include more details regarding these calculations in the text, and an economic interpretation of equations (3) and (4) should be provided as well. Next, the detailed calculations should also be appended in the supplementary section for the readers who want to replicate your results.

If there is enough space, equation (18) in Hendricks and McAfee (2010) should also be included, as it clearly shows how sellers understate their biomass in the report mechanism. The calculation of the Nash equilibrium turns out to be central in the construction of the (interpolated) access policy functions described on page 21.

In general, I think that the authors rely too much on the reader being

familiar with Hendricks and McAfee (2010). While the latter is a good (perhaps underrated) paper, it is arguably not a ‘classic’ of the IO literature. The reader, who does not know it, should be able to read your paper without referring to it constantly.

Minor points:

- On page 10, you mention that the economic surplus that you compute in your model simulations does not include “the economic value of increased no-access catch.” However, on page 13, when you introduce the opportunity cost, $c(\cdot)$, of supplying access, you write that it captures “the value of the (foregone) potential domestic catch” which would be non-access catch, as per your definition of non-access catch on page 5 (second paragraph). Is there a contradiction here? I might be wrong, but I am wondering if you should not better clarify what the cost function $c(\cdot)$ stands for.
- A list of selling countries should be provided (for the preferred threshold). On the maps (e.g. figure 2), the authors report changes for every coastal country from Africa. It seems that the island “La Réunion” which is located east of Madagascar is also featured on these maps (I may have missed it but I was not able to find the whole list of the selling countries in the “supplementary materials”). The island of La Réunion is not an independent African country (or even autonomous region) since it is a ‘departement’ of France (like say “Calvados”). Thus, technically, it is part of the EU who is also a buying country. Does this island negotiate an access fee with the EU? If it is part of the list, should it be removed from the selling countries for correctness?
- The variable “gt”. It seems a bit unusual and heavy to use two letters for a (single) variable. Does ‘net tonnage’ plays an important role in the paper? If not, perhaps the letter t for “gross tonnage” would perhaps be sufficient. This is a matter of taste however.
- On page 15, you wrote: “Our results accord this theorem.” Is *with* missing?
- On page 19, second paragraph, you wrote: “Each country’s stock stay within their original, status quo boundaries in the coalition scenario.”

The precise meaning of this sentence eludes me simply because the biomass ends up increasing substantially with the coalition equilibrium. What do you mean exactly? Do you mean that the growth curve remains unchanged despite switching to the coalition scenario? Please explain in the text.

- In the supplementary material (p. 27-28). The table S8 and S9 mentions “African sellers”. That does not sound right since you are dealing with the PNA countries.

August 14, 2023

NCOMMS-23-23802: “A fish cartel for Africa”

Dear Referee 1,

Thank you very much for your thoughtful comments on our paper. We have revised our paper accordingly, and we are grateful for the opportunity to resubmit it. We provide a point-by-point letter in response to your excellent comments in this document.

We reproduce your comments in bold face, respond to your comments in plain text, *and indicate relevant text from our revised paper with italics.*

This is an interesting paper addressing an important issue in terms of natural resource extraction in developing countries. The paper is generally well motivated and executed. However, there is one part of the empirical outcome that warrants more discussion. The result that by reducing the access of foreign fishers will lead to both higher total landings and larger fish stocks seems counter-intuitive and almost too good to be true as most African nations do not manage their fisheries very effectively.

We are glad you like our paper. We address your specific questions and concerns related to higher total landings and larger fish stocks throughout the remainder of our letter.

Ln 3. I do not like the start of the abstract. It assumes that the reader knows a lot about the trade with fishing rights. It would be good if this could be reformulated to make it more motivational.

We have revised the abstract by adding a new first sentence:

Many countries sell fishing rights to foreign nations and fishers. Although African coastal waters are among the world’s most biologically rich, African countries earn much less than their peers from selling access to foreign fishers. African countries sell fishing access individually (in contrast to some Pacific countries who sell access as a bloc). We develop a bilateral

oligopoly model to simulate the effects of an African fish cartel. The model shows that wielding market power entails both ecological and economic dimensions. Africa would substantially restrict access catch, which raises biomass by 16%. But this also confers economic benefits to all African nations, raising profits by an average of 23%. These benefits arise because market power shifts from foreign buyers to African sellers. While impediments to sustainable development like corruption are hard to change in the medium-term, deeper African integration is an already-emerging solution to African countries' economic and ecological challenges.

We believe the new first sentence helps introduce the topic for readers who are not familiar with the trade in fishing rights. We would have liked to add more contextual and motivational information to the abstract, but we are constrained by *Nature Communication's* abstract limit of 150 words. Our original abstract was 138 words, and the revised abstract is 150 words.

Ln 18. While former colonizers are important in this trade, I think you should be explicit that there are also other agents with China probably being the most important today, and countries like South Korea and Taiwan are important. Until its collapse, also the USSR and other eastern block countries were important distant water fishing nations that bought or bartered fishing rights.

We have revised the fourth sentence of the paper to explicitly note the important role of China, Taiwan, and South Korea:

Fisheries may be the most poignant example; newly available satellite data reveal that foreign vessels, including those from more recently-industrialized China, Taiwan, and South Korea, comprise more than half of fishing activity in African waters and pay pennies on the dollar for access to these waters (2, 3).

Ln 27. It would be good if you mentioned a few examples here before you focus on fish.

We have revised the sentence to provide three examples before we narrow the scope of the study to fish:

While these questions may apply across a wide range of natural resources, such as critical minerals, oil, and natural gas, we study the case of international fishing access agreements.

Ln 48. Would it not also make sense to put in management here? As I understand it, the PNA countries have a management system that is doing better in protecting stocks and thereby presumably increasing CPUE.

Yes, we have revised the text to explicitly include management as a possible difference between African and PNA countries:

Similarly, if African fish stocks are depleted due to inferior fisheries management, are more likely to cross international boundaries, are composed of species that command lower market prices, or must be transported longer distances to final markets, then foreign willingness to pay to fish in African waters will also be lower (8).

Ln 73. Fang and Asche (2021, Ecological Economics) provides evidence that the U.S. have buyer power for several species.

Thank you for this helpful citation. We have added it to the end of the relevant sentence:

What complicates this story is that there is also market power on the demand side (10).

Ln 91. I think you should provide some more intuition for the outcomes here even though some of it is explained in more detail in the next section. I think it is particularly important to note that if the cartel is to exploit market power, the supply will be reduced and therefore the stock status improved (if the foreign catches are not completely crowded out by higher domestic landings), and as such that there is also likely to be a conservation benefit associated with the cartel.

We have revised the paragraph to include a new final sentence:

We compare two scenarios: the “status quo” scenario in which African countries sell access as individual countries, and the “coalition” scenario in which African countries sell access as a bloc. This institutional difference is the only difference between the two scenarios; everything else is the same, such as enforcement against illegal fishing, legal fishing costs, and the degree of fish stock movement between national and international waters. We are therefore able to isolate the effects of enhanced African market power on profits, catch, and fish stocks, holding other parameters and institutions fixed. If African countries use their

enhanced market power to restrict the supply of fishing access, profits to African countries and the health of African fish stocks could improve.

We believe this new final sentence provides the intuition you request in advance of the more detailed discussion in the immediately following paragraph.

You do not mention prices here. I think you should explicitly note that you regard the price of the fish as exogenous as there is a global market for most species (see e.g. Anderson et al., 2018, Journal of Commodity Markets).

We have revised the relevant sentence to make the explicit note you request, and we have cited Anderson et al. (2018) to support our assumption:

We are therefore able to isolate the effects of enhanced African market power on profits, catch, and fish stocks, holding other institutions and parameters such as fish price fixed (12).

Ln 145. I am a bit surprised at this result that total domestic landings increase more than the reduction in foreign landings. Can you please provide more intuition to explain this result as that would lead to a lower CPUE for domestic fishers and presumably lower profits.

We have made two revisions to the paragraph to provide more intuition for the result.

First, we added the sentence:

The 16% increase in biomass is why non-access catch can more than compensate for the reduction in access catch.

Thus, CPUE and profits for domestic and unauthorized foreign fishers need not be lower in the Africa Coalition.

Second, we revised one sentence to remind readers that non-access catch includes both domestic catch and unauthorized foreign catch:

Non-access catch, which includes both domestic and unauthorized foreign catch, increases from 9.46 million tons to 10.36 million tons per year.

Ln 148. And I really struggle to understand the result that fish stocks increase even with higher landings. That would be true if African fisheries are on the forward part of the backward-bending supply schedule, but that seems odd given the lack of management. I assume this follows from you partially depleted assumption (ln 342), but is that not overly optimistic? Is 0.6 sufficient as a robustness check? For me it seems like that is also too optimistic as it still appears as you are not on the backward bending part of the supply schedule.

We added an additional robustness check to assess the sensitivity of our results to assuming $\frac{b}{b_{MSY}}$ equals 0.8 or 0.6. Instead of assuming $\frac{b}{b_{MSY}}$ is 0.8 or even 0.6, we re-ran our model assuming $\frac{b}{b_{MSY}} = 0.4$.

As an additional robustness check, we repeat our analysis assuming $\frac{b}{b_{MSY}} = 0.4$. In this case, we estimate that a coalition would increase total biomass in African waters by 25 million tons and total seller profits by \$48 million (Table S3, Figure S7, and Figure S8).

These gains in biomass and seller profits from the Africa Coalition are similar in magnitude to the gains we calculate in our primary specification of $\frac{b}{b_{MSY}} = 0.8$. In our primary specification, we estimate that a coalition would increase total biomass in African waters by 19 million tons and total seller profits by \$37 million. We attribute the slightly larger gains in the $\frac{b}{b_{MSY}} = 0.4$ scenario to the increased potential for African fish stocks to recover from a more depleted state.

	Status quo	Coalition	Difference	% Difference
Catch, access (millions of tons)	2.47	1.93	-0.54	-22%
Access fee per ton	\$128.20	\$146.05	\$17.85	14%
African sellers' profit (millions)	\$162.15	\$210.26	\$48.11	30%
Foreign buyers' profit (millions)	\$363.15	\$323.44	-\$39.71	-11%
Catch, non-access (millions of tons)	9.46	11.15	1.69	18%
Catch, total (millions of tons)	11.93	13.08	1.16	10%
Biomass (millions of tons)	77.54	102.13	24.59	32%

Table S3: **Effect of Africa Coalition on catch, profit, and biomass when $\frac{b}{b_{MSY}} = 0.4$.** Catch and profit are annual quantities. Access fee and profit are in 2020 USD. In our baseline specification we calculate each selling country's true status quo biomass with the assumption that $\frac{b}{b_{MSY}} = 0.8$ (Methods). As a robustness check, we set $\frac{b}{b_{MSY}} = 0.4$ and then we repeat our analysis.

Figure S7: **Effect of Africa Coalition on access catch and profit by selling country when $\frac{b}{b_{MSY}} = 0.4$.** Percent change is relative to status quo value. In our baseline specification we calculate each selling country's true status quo biomass with the assumption that $\frac{b}{b_{MSY}} = 0.8$ (Methods). As a robustness check, we set $\frac{b}{b_{MSY}} = 0.4$ and then we repeat our analysis.

Figure S8: **Effect of Africa Coalition on total catch and biomass by selling country when $\frac{b}{b_{MSY}} = 0.4$.** Percent change is relative to status quo value. In our baseline specification we calculate each selling country's true status quo biomass with the assumption that $\frac{b}{b_{MSY}} = 0.8$ (Methods). As a robustness check, we set $\frac{b}{b_{MSY}} = 0.4$ and then we repeat our analysis.

August 14, 2023

NCOMMS-23-23802: “A fish cartel for Africa”

Dear Professor Marcoul,

Thank you very much for your encouraging and helpful comments on our paper. We have revised our paper accordingly, and we are grateful for the opportunity to resubmit it. We provide a point-by-point letter in response to your excellent comments in this document.

We reproduce your comments in bold face, respond to your comments in plain text, *and indicate relevant text from our revised paper with italics.*

Background and policy questions: The economic problem analyzed in this work revolves around the sale of fishing access rights to their waters by African countries. Currently, all African countries individually negotiate fishing access with foreign countries (or a block of foreign countries), such as the European Union. These agreements usually specify an allowable tonnage for different species and an access fee per ton of fish caught in the Exclusive Economic Zone of the signing country. Moreover, it is notorious that these agreements are quite unfavorable to African countries, leaving them only with a tiny fraction of the value of the fish caught.

The paper’s main question is simple: What would happen if African countries were cooperating and selling access rights as a cartel rather than separately?

The analysis conducted shows that if such a cartel was formed, African countries would widely benefit not only through a steep increase of profit earned (from the sale of access rights), but also, through a higher biomass in their waters.

In a related question, they also analyze the existing fish cartel formed by the Pacific Island countries (PNA). Taking an opposite view, they ask what would happen if the PNA cartel was dismantled and reverted to a situation where each Pacific island country was individually negotiating access rights with buying countries. The answer to this question shows that PNA countries have greatly benefited from the cartel formation.

Evaluation: I found this paper and its results highly relevant for three rea-

sons. First, the idea and the contribution of the paper are novel and original. I am not aware of any comparable studies in the renewable resource literature. Second, the quantified evaluation that the authors can provide aligns with my economic expectations; I found these findings to be credible. My impression of credibility is reinforced by the methods used, which are both rigorous and well-executed. Last but not least, the subject of the study is highly topical and has far-reaching consequences. It will be of interest not just to resource and development economists but also to development practitioners and African fishery decision-makers of coastal countries. With the quantified evaluations contained in the paper, the latter could find a solid starting point to engage in a conversation with each other.

Below, I develop a few comments/points that the authors may need to consider in revising the paper.

- The idea that non-access catch increases as a result of the coalition can, perhaps non-intuitively, be a good consequence of the coalition. This will be especially true if non-access catches are mostly made by subsistence fishers (or at least relatively poor ones). It would appear to me that if there is less access catch for foreign fishers, then local artisan fishers would be better off as a result of less competition on the fishing grounds. Hence, African countries would receive a benefit in the form of larger fish consumption for coastal communities (hardly a bad thing), and not necessarily in the form of a larger resource for the central state. If a poor state is subject to a high degree of resource misappropriation (e.g., corruption), then the former might be preferable to the latter. The authors should acknowledge this aspect, and perhaps develop it. Note that if the authors are to develop the discussion in this direction, then, it would be useful to report the change in non-access catch in Figure 2 of page 9.

We appreciate you highlighting the benefits of greater non-access catch, and we agree that it significantly contributes to the overall impacts of the proposed Africa Coalition. We have expanded our discussion of the increase in non-access catch with the following revisions. First, we added the following text to the Discussion:

For instance, 48% of non-access catch in African sellers' waters is domestic, rather than unauthorized foreign (13). Increased domestic catch could lead to larger fish consumption for coastal communities, and less competition for fishing grounds from foreign vessels operating

Figure S18: **Effect of Africa Coalition on non-access catch.** (a) Status quo values and (b) percent change in non-access catch under Africa Coalition relative to status quo values.

under access agreements could result in higher profits for domestic fishers.

Second, we added a figure displaying the effect of the Africa Coalition on non-access catch, and we reference the new figure in the Results section:

Total catch slightly increases for all countries, as the increase in non-access catch more than compensates for the decrease in access catch (Figure S18).

We placed the figure in the Appendix, rather than in Figure 2, because we hope to avoid shrinking the four subfigures of Figure 2.

- **Related to the previous point.** I am concerned by the assumption made on page 6 , where it is assumed that countries’ non-access policy functions “are the same in both scenarios.” By switching to a cartel, it seems intuitive that smallholder fishers would compete (much) less harshly with foreign (international) fishers, who are usually better equipped. As such, the non-

access catch policy function would change (not in a linear fashion). If the shape of the non-access catch policy function is different after the coalition is formed, then it is less clear how the benefits received by the African countries may change.

We have expanded our discussion of how the benefits of an Africa Coalition would change if the non-access policy function changes:

We assume countries' non-access policy functions are the same in both scenarios; while increases in biomass are still met with an increase in non-access catch, the function itself is unchanged as a result of the coalition. In fact, it is plausible that at any given level of biomass, there would actually be less non-access catch in an Africa Coalition (in other words, the coalition could cause the non-access policy function to pivot down). For example, if the coalition makes countries more able to deter illegal fishing, then the gain in biomass would be even larger than our estimates. But if the opposite occurs, then our estimates would overstate the gain in biomass. For example, if domestic fishers' catch increases at any given biomass level due to reduced competition from foreign vessels operating under access agreements, then the gain in biomass from the Africa Coalition would be smaller than our estimates.

- **Regarding the Africa Versus PNA coalition.** An important observation is that the PNA, as analyzed by the authors, seems to be a Pareto optimum (see p.11). The PNA is thus stable in the sense that no player within this agreement has any incentives to break it. For instance, it would be counter-productive for buying countries to attempt to weaken the bargaining coalition formed by the (selling) Pacific Island Nations. In fact, buying countries may even subsidize the administrative costs of organizing the coalition of selling countries. Is there any evidence/hint that buying countries have indeed favored the formation of the PNA cartel? If yes, this information should be reported as corroboration (or weak evidence) that the approach and calculations of the authors are correct.

In the case of Africa, this particular configuration is not present and sellers and buyers have opposite interests (even if total surplus is strictly larger). Buying countries may thus insist on dealing with an individual selling country.

We have added the following paragraph to the Methods section:

In our main text discussion, we noted that buying countries benefit from the PNA in the sense that their profit would be lower if the PNA did not exist. This result raises the question of whether buying countries explicitly supported the creation of the PNA. While we could not find documentary evidence, our sense is that buying countries such as China, Japan, and Taiwan display partial support for the PNA's existence. In the short-run, buying countries may see the PNA as rent-decreasing, owing to the market power it allows selling countries to wield in the pricing of fishing access. With the long-run in mind, however, the PNA is regarded as having been instrumental in helping to ensure sustainable fishery management for most tuna stocks in the Pacific.

- **Report of fishing capacity (p. 14).** How is the gross tonnage of country j , gt_j , determined? Does the auctioneer/market mechanism equates the following equation for country j

$$\frac{gt_j}{\sum_{j' \in J} gt_{j'}} = \frac{\widehat{gt}_j}{\sum_{j' \in J} \widehat{gt}_{j'}}?$$

I assumed it was a similar process to the one used for determining the quantity share q_i of the selling countries. Unless I missed it, I think this equation should be mentioned in the text.

You are correct that the market mechanism determines buyers' quantity shares in the same manner that it determines sellers' quantity shares. We have added the following sentence and equation to the text:

The market mechanism sets j 's quantity share equal to their share of total reported gross tonnage: $q_j = \frac{\hat{t}_j}{\hat{T}} Q(\hat{B}, \hat{T})$.

- **Determination of the equilibrium price (p.13-15).** I did not find the determination of equations (3) and (4) easy. My understanding is that to do so, one must compute the supply functions of the selling countries and the demand for access permits of the buying countries, but also, one must use the two equations of the market mechanism (i.e., those linking reports with

biomass and permit allocations). I would suggest that you include more details regarding these calculations in the text, and an economic interpretation of equations (3) and (4) should be provided as well. Next, the detailed calculations should also be appended in the supplementary section for the readers who want to replicate your results.

We have added the following details regarding the calculation of Equations 3 and 4 to the Methods section, as well as an economic interpretation of the equations:

The equilibrium price decreases in reported supply (\hat{B}) and increases in reported demand (\hat{T}), while the equilibrium quantity increases in both reported supply and reported demand. The equilibrium price and quantity occur from equating the partial derivatives of opportunity cost and fishing profit (Supplementary derivations). The market mechanism therefore finds the price and quantity that maximizes total economic surplus assuming sellers and buyers' reports are truthful. The market mechanism is efficient conditional on reports, but when reports differ from true biomass or gross tonnage values, the resulting equilibrium will not maximize total economic surplus.

We have added the following supplementary text to detail the calculations:

Supplementary derivations

In Hendricks and McAfee (2010), the market mechanism solves for the quantity that equates the marginal opportunity cost with the marginal fishing profit, both of which are evaluated at the market-level values. Re-writing their Equation 8 in the language of our model, and letting ' indicate the partial derivative operator, we have

$$c'(\frac{Q(\hat{B}, \hat{T})}{\hat{B}}) = v'(\frac{Q(\hat{B}, \hat{T})}{\hat{T}})$$

In their constant elasticities special case, the marginal opportunity cost and marginal fishing profit take the forms

$$(\frac{Q(\hat{B}, \hat{T})}{\hat{B}})^{1/\eta} = (\frac{Q(\hat{B}, \hat{T})}{\hat{T}})^{-1/\epsilon} \quad (11)$$

(bottom right column of page 397). We can re-arrange Equation 11 to solve for the equilibrium quantity, which we stated in Equation 4:

$$\begin{aligned}
\frac{Q(\hat{B}, \hat{T})}{\hat{B}} &= \left(\frac{Q(\hat{B}, \hat{T})}{\hat{T}}\right)^{-\eta/\epsilon} \\
\left(\frac{Q(\hat{B}, \hat{T})}{\hat{B}}\right)^\epsilon &= \left(\frac{Q(\hat{B}, \hat{T})}{\hat{T}}\right)^{-\eta} \\
\left(\frac{Q(\hat{B}, \hat{T})}{\hat{B}}\right)^\epsilon &= \left(\frac{\hat{T}}{Q(\hat{B}, \hat{T})}\right)^\eta \\
Q(\hat{B}, \hat{T})^{\epsilon+\eta} &= \hat{B}^\epsilon \hat{T}^\eta \Rightarrow \\
Q(\hat{B}, \hat{T}) &= \hat{B}^{\epsilon/(\epsilon+\eta)} \hat{T}^{\eta/(\epsilon+\eta)}
\end{aligned}$$

The equilibrium price, which we stated in Equation 3, occurs when the marginal opportunity cost equals the marginal fishing price for all sellers and buyers. Re-writing their Equation 5 in the language of our model, we have

$$c'\left(\frac{q_i}{\hat{b}_i}\right) = p = v'\left(\frac{q_j}{\hat{t}_j}\right) \quad \forall i, j \quad (12)$$

We solve for the equilibrium price in the constant elasticities case by re-arranging the left side of Equation 12, substituting $\frac{\hat{b}_i Q(\hat{B}, \hat{T})}{\hat{B}}$ for q_i , and plugging in the equilibrium quantity:

$$\begin{aligned}
c'\left(\frac{\hat{b}_i Q(\hat{B}, \hat{T})}{\hat{B} \hat{b}_i}\right) &= p(\hat{B}, \hat{T}) \\
c'\left(\frac{Q(\hat{B}, \hat{T})}{\hat{B}}\right) &= p(\hat{B}, \hat{T}) \\
\left(\frac{Q(\hat{B}, \hat{T})}{\hat{B}}\right)^{1/\eta} &= p(\hat{B}, \hat{T}) \\
\left(\frac{\hat{B}^{\epsilon/(\epsilon+\eta)} \hat{T}^{\eta/(\epsilon+\eta)}}{\hat{B}}\right)^{1/\eta} &= p(\hat{B}, \hat{T}) \\
\left(\frac{\hat{T}^{\eta/(\epsilon+\eta)}}{\hat{B}^{\eta/(\epsilon+\eta)}}\right)^{1/\eta} &= p(\hat{B}, \hat{T}) \Rightarrow \\
p(\hat{B}, \hat{T}) &= \hat{B}^{-1/(\epsilon+\eta)} \hat{T}^{1/(\epsilon+\eta)}
\end{aligned}$$

If there is enough space, equation (18) in Hendricks and McAfee (2010) should also be included, as it clearly shows how sellers understate their

biomass in the report mechanism. The calculation of the Nash equilibrium turns out to be central in the construction of the (interpolated) access policy functions described on page 21.

In general, I think that the authors rely too much on the reader being familiar with Hendricks and McAfee (2010). While the latter is a good (perhaps underrated) paper, it is arguably not a 'classic' of the IO literature. The reader, who does not know it, should be able to read your paper without referring to it constantly.

We have added Equation 18 from Hendricks and McAfee (2010) to the Methods section, as well as an interpretation of the equation:

Equation 18 in Hendricks and McAfee (2010) expresses the relationship between reports and true values. Re-writing that equation in the language of our model, we have

$$\begin{aligned}\frac{\hat{b}_i}{b_i} &= \left(1 - \frac{\sigma_i}{\epsilon + \eta \cdot (1 - \sigma_i)}\right)^\eta \\ \frac{\hat{t}_j}{t_j} &= \left(1 + \frac{s_j}{\epsilon \cdot (1 - s_j) + \eta}\right)^{-\epsilon}\end{aligned}\tag{5}$$

where $\sigma_i = \frac{\hat{b}_i}{B}$ and $s_j = \frac{\hat{t}_j}{T_j}$. Writing Equation 5 in terms of quantity shares σ_i and s_j provides intuition regarding the market power mechanism in our model. The larger the share of the market a seller or buyer represents, the more they will understate their biomass or gross tonnage capacity.

We re-arrange Equation 5 to solve for Seller i 's equilibrium report...

Minor points:

- On page 10, you mention that the economic surplus that you compute in your model simulations does not include “the economic value of increased no-access catch.” However, on page 13, when you introduce the opportunity cost, $c(\cdot)$, of supplying access, you write that it captures “the value of the (foregone) potential domestic catch” which would be non-access catch, as per your definition of non-access catch on page 5 (second paragraph). Is there a contradiction here? I might be wrong, but I am wondering if you

should not better clarify what the cost function $c(\cdot)$ stands for.

You are correct. We have removed the contradictory clause you identified (“the value of the (foregone) potential domestic catch”), and we have clarified that $c(\cdot)$ is (solely) the opportunity cost or shadow cost of supplying access to foreign fishers:

Selling Country i 's opportunity cost (or shadow cost) of supplying access, $c(\cdot)$, captures the value of the future stock growth that i foregoes in supplying access to foreign fishers.

- **A list of selling countries should be provided (for the preferred threshold). On the maps (e.g. figure 2), the authors report changes for every coastal country from Africa. It seems that the island “La Réunion” which is located east of Madagascar is also featured on these maps (I may have missed it but I was not able to find the whole list of the selling countries in the “supplementary materials”). The island of La Réunion is not an independent African country (or even autonomous region) since it is a ‘departement’ of France (like say “Calvados”). Thus, technically, it is part of the EU who is also a buying country. Does this island negotiate an access fee with the EU? If it is part of the list, should it be removed from the selling countries for correctness?**

We apologize for the confusion: La Réunion is not included in our paper’s figures. Below we plot the Exclusive Economic Zones of our selling countries in blue, and the Exclusive Economic Zone of La Réunion in red.

La Réunion added in red

We have added a list of selling countries as Figure S2b:

Figure S2: (a) **Gross tonnage of each buying country's distant-water fishing (DWF) fleet and (b) True status quo biomass of each selling country.** The x-axes display the ISO3 code of each country. We model the European Union (EU) as a single country because the EU purchases fishing access as a bloc.

- The variable “gt”. It seems a bit unusual and heavy to use two letters for a (single) variable. Does ‘net tonnage’ play an important role in the paper? If not, perhaps the letter t for “gross tonnage” would perhaps be sufficient. This is a matter of taste however.

We have changed all instances of “gt” to “t”.

- On page 15, you wrote: “Our results accord this theorem.” Is with missing?

We have added the word “with”.

- On page 19, second paragraph, you wrote: “Each country’s stock stay within their original, status quo boundaries in the coalition scenario.” The precise meaning of this sentence eludes me simply because the biomass ends up increasing substantially with the coalition equilibrium. What do you mean exactly? Do you mean that the growth curve remains unchanged despite switching to the coalition scenario? Please explain in the text.

Your understanding is correct. We have added your correct interpretation to the paragraph to clarify our meaning:

Moreover, the growth curve for each country’s stock remains unchanged in the coalition scenario.

- In the supplementary material (p. 27-28). The table S8 and S9 mentions “African sellers”. That does not sound right since you are dealing with the PNA countries.

Thank you for noticing these typos. We have corrected them.

Reviewers' Comments:

Reviewer #1:

Remarks to the Author:

Thanks for doing a good job at addressing my concerns and for an interesting paper.

Reviewer #2:

Remarks to the Author:

Dear authors,

I am satisfied with your revisions of the manuscript.

Philippe Marcoul

October 7, 2023

NCOMMS-23-23802: “A fish cartel for Africa”

Dear Referee 1,

We reproduce your letter in bold face and respond to your letter in plain text.

Thanks for doing a good job at addressing my concerns and for an interesting paper.

We are pleased you are satisfied with our revisions. Thank you again for reviewing our paper.

Sincerely,

Gabriel Englander and Christopher Costello

October 7, 2023

NCOMMS-23-23802: “A fish cartel for Africa”

Dear Professor Marcoul (Referee 2),

We reproduce your letter in bold face and respond to your letter in plain text.

I am satisfied with your revisions of the manuscript.

We are pleased you are satisfied with our revisions. Thank you again for reviewing our paper.

Sincerely,

Gabriel Englander and Christopher Costello